# Albiflorin Decreases Glutamate Release from Rat Cerebral Cortex Nerve Terminals (Synaptosomes) through Depressing P/Q-Type Calcium Channels and Protein Kinase A Activity

**DOI:** 10.3390/ijms25168846

**Published:** 2024-08-14

**Authors:** Cheng-Wei Lu, Tzu-Yu Lin, Ya-Ying Chang, Kuan-Ming Chiu, Ming-Yi Lee, Su-Jane Wang

**Affiliations:** 1Department of Anesthesiology, Far-Eastern Memorial Hospital, New Taipei City 22060, Taiwan; drluchengwei@gmail.com (C.-W.L.); drlin1971@gmail.com (T.-Y.L.); yychang0310@saturn.yzu.edu.tw (Y.-Y.C.); 2Department of Mechanical Engineering, Yuan Ze University, Taoyuan 32003, Taiwan; 3International Program in Engineering for Bachelor, Yuan Ze University, Taoyuan 32003, Taiwan; 4Division of Cardiovascular Surgery, Cardiovascular Center, Far-Eastern Memorial Hospital, New Taipei City 22060, Taiwan; chiu9101018@gmail.com; 5Department of Electrical Engineering, Yuan Ze University, Taoyuan 32003, Taiwan; 6Department of Medical Research, Far-Eastern Memorial Hospital, New Taipei City 22060, Taiwan; mingyi.lee@gmail.com; 7School of Medicine, Fu Jen Catholic University, New Taipei City 24205, Taiwan; 8Research Center for Chinese Herbal Medicine, College of Human Ecology, Chang Gung University of Science and Technology, Taoyuan 33303, Taiwan

**Keywords:** albiflorin, glutamate release inhibition, Ca^2+^ channel, PKA, synaptosome

## Abstract

The purpose of this study was to investigate whether and how albiflorin, a natural monoterpene glycoside, affects the release of glutamate, one of the most important neurotransmitters involved in neurotoxicity, from cerebrocortical nerve terminals (synaptosomes) in rats. The results showed that albiflorin reduced 4-aminopyridine (4-AP)-elicited glutamate release from synaptosomes, which was abrogated in the absence of extracellular Ca^2+^ or in the presence of the vesicular glutamate transporter inhibitor or a P/Q-type Ca^2+^ channel inhibitor, indicating a mechanism of action involving Ca^2+^-dependent depression of vesicular exocytotic glutamate release. Albiflorin failed to alter the increase in the fluorescence intensity of 3,3-diethylthiacarbocyanine iodide (DiSC_3_(5)), a membrane-potential-sensitive dye. In addition, the suppression of protein kinase A (PKA) abolished the effect of albiflorin on glutamate release. Albiflorin also reduced the phosphorylation of PKA and synaptosomal-associated protein of 25 kDa (SNAP-25) and synapsin I at PKA-specific residues, which correlated with decreased available synaptic vesicles. The results of transmission electron microscopy (TEM) also observed that albiflorin reduces the release competence of synaptic vesicles evoked by 4-AP in synaptosomes. In conclusion, by studying synaptosomally released glutamate, we suggested that albiflorin reduces vesicular exocytotic glutamate release by decreasing extracellular Ca^2+^ entry via P/Q-type Ca^2+^ channels and reducing PKA-mediated synapsin I and SNAP-25 phosphorylation.

## 1. Introduction

The neurotransmitter glutamate has an important role in excitatory neurotransmission in the mammalian central nervous system (CNS), but excessive glutamate leads to excitotoxicity. This event is present in several neurological disorders, such as epilepsy, traumatic brain injury, stroke, and Alzheimer’s disease [1]. A high concentration of glutamate in the synaptic cleft leads to the overstimulation of glutamate receptors, which in turn leads to increased calcium levels, mitochondrial abnormalities, oxidative stress, and eventually cell atrophy and death [1]. Therefore, a normal level of synaptic glutamate is necessary for the prevention of excitotoxicity, and this may potentially be achieved through the regulation of synaptic glutamate release [2].

Natural chemicals from food or herbs play a crucial role in the development of safe and effective treatments for CNS disease therapy. Albiflorin (C23H28O11, MW 480.5; Figure 1A), a monoterpene glycoside, is an active ingredient of the roots of *Paeonia albiflora* that is used in traditional Chinese medicine and has medicinal properties [3,4]. Albiflorin has antioxidant, anti-inflammatory, immunoregulatory, analgesic, as well as neuroprotective effects, without causing overt toxicity [5,6,7,8]. Regarding its neuroprotective effects, albiflorin has been demonstrated to reduce neuronal loss, attenuate brain damage, and alleviate mood and memory deficits in several experimental models of neurological disorders, including spinal cord injury, cerebral ischemia, depression, post-traumatic stress disorder, and Alzheimer’s disease [9,10,11,12,13,14,15]. However, to the best of our knowledge, no research has examined the effect of albiflorin on the regulation of synaptic glutamate release, which is a crucial mechanism for neuroprotective action [2,16,17]. Therefore, the aim of this study was to examine the effect of albiflorin on the release of glutamate and the underlying mechanism using rat cerebral cortex nerve terminals (synaptosomes). Synaptosomes can accumulate, store, and release neurotransmitters and do not suffer from any postsynaptic interactions. Therefore, the preparation and modulation of synaptosomes is a well-established in vitro technique for investigating synaptic neurotransmitter release [18].

## 2. Results

### 2.1. Albiflorin Reduced Ca^2+^-Dependent Vesicular Exocytotic Glutamate Release from Cerebral Cortical Synaptosomes in Rats

Experiments were conducted to investigate whether albiflorin can modify 4-AP (1 mM)-elicited glutamate release from cerebral cortical synaptosomes. 4-AP is a K^+^ channel blocker that mimics the physiological mechanisms of terminal depolarization and the Ca^2+^-dependent vesicular exocytotic glutamate release [19]. As shown in Figure 1B, the exposure of synaptosomes to 4-AP (1 mM) elicited glutamate overflow of 7.6 ± 0.4 nmol/mg/5 min. Glutamate efflux from synaptosomes elicited by 4-AP was lower when the synaptosomes were preincubated with 10 μM albiflorin for 10 min than when they were treated with 4-AP alone [t(8) = 23.1, *p* < 0.0001]. Albiflorin did not alter the basal, pre-depolarization glutamate level (*p* > 0.05). The effect of albiflorin was dose-dependent, with 3, 5, 10, and 30 μM albiflorin depressing glutamate release to 86% (*p* < 0.0001), 74% (*p* < 0.0001), 52% (*p* < 0.0001), and 50% (*p* < 0.0001) of the control values, respectively (Figure 1C). The IC_50_ value was 18 μM. In addition, the effect of albiflorin was efficiently prevented by the concomitant presence of bafilomycin A1 [F(2,12) = 355.6, *p* < 0.0001], an inhibitor of vesicular glutamate transporters, which blocks the uptake of glutamate into synaptic vesicles, as well as by the application of EGTA to synaptosomes (in the extracellular Ca^2+^-free medium) [F(2,12) = 470.5, *p* < 0.0001] (Figure 1D). Bafilomycin A1 and Ca^2+^-free medium significantly affected glutamate efflux elicited by 4-AP (*p* < 0.0001; Figure 1D). However, no statistical difference was observed between the release after bafilomycin A1 or Ca^2+^-free medium alone and after bafilomycin A1 + albiflorin or Ca^2+^-free medium + albiflorin treatment (*p* > 0.05).

### 2.2. Reduced Glutamate Release from Albiflorin Is Mediated through Ca^2+^ Channel Suppression

The ability of albiflorin to depress glutamate overflow from synaptosomes was further assessed. As shown in Figure 2, the release of glutamate elicited by 4-AP was greatly reduced by ω-conotoxin GVIA, an N-type Ca^2+^ channel blocker [F(2,12) = 256.7, *p* < 0.0001], or ω-agatoxin IVA, a P/Q-type Ca^2+^ channel blocker [F(2,12) = 233.3, *p* < 0.0001]. In the presence of ω-conotoxin GVIA, albiflorin continued to significantly reduce the 4-AP-induced release of glutamate (*p* < 0.0001). In contrast to the ω-conotoxin GVIA, the effect of albiflorin was prevented by the ω-agatoxin IVA pretreatment, and no significant difference was observed between the glutamate release after ω-agatoxin IVA treatment alone and after the ω-agatoxin IVA and albiflorin treatment (*p* = 0.8).

### 2.3. Albiflorin Failed to Affect the Synaptosomal Membrane Potential

The suppression of Ca^2+^ channels by albiflorin might be due to an alteration in the synaptosomal plasma membrane potential, which consequently modulates Ca^2+^ influx into the terminal. Therefore, the effect of albiflorin on the synaptosomal plasma membrane potential was assessed with the membrane-potential-sensitive dye DiSC_3_(5). As shown in Figure 3, 4-AP (1 mM) increased the fluorescence intensity of DiSC_3_(5). Compared with that of 4-AP alone, the presence of albiflorin did not significantly affect the increase in DiSC_3_(5) fluorescence intensity caused by 4-AP [t(8) = 0.08, *p* = 0.94].

### 2.4. The Albiflorin-Mediated Inhibition of Glutamate Release Involved the Suppression of the Protein Kinase A (PKA) Pathway

One of the important modulators of vesicular exocytotic glutamate release is presynaptic PKA [20]. As shown in Figure 4A, the PKA inhibitor H89 significantly reduced the glutamate release elicited by 4-AP [F(2,12) = 132.8, *p* < 0.0001]. The effect of albiflorin was efficiently prevented by the concomitant presence of H89, there being no statistical difference between the release after H89 alone and after albiflorin + H89 treatment (*p* = 0.94). Similar results were observed with another PKA inhibitor, KT5720 (1 μM) [F(2,12) = 100.4, *p* < 0.0001]. Albiflorin only reduced glutamate release by an additional 1% in synaptosome pretreated with KT5720 (*p* = 0.98). Furthermore, 4-AP (1 mM) increased the phosphorylation of PKA [F(2,12) = 632.7, *p* < 0.0001], which was also markedly decreased after exposure to albiflorin (*p* < 0.001). The levels of PKA were not significantly different among the groups [F(2,12) = 0.003, *p* = 1] (Figure 4B–D).

To further explore the regulatory effect of albiflorin on presynaptic PKA pathway, we assessed the phosphorylation of SNAP-25 and synapsin 1, which are presynaptic substrates for PKA [19]. PKA phosphorylates SNAP-25 on threonine 138 (T138) and synapsin I on serine 9 (S9), promoting the trafficking and mobilization of synaptic vesicles to nerve terminals and consequently increasing the size of the releasable vesicle pools [21,22]. As shown in Figure 5A–E, compared to the control, 1 mM 4-AP increased the levels of p-138-SNAP-25 [F(2,12) = 347.3, *p* < 0.0001] and p-S9-synapsin I [F(2,12) = 139.4, *p* < 0.0001] without altering the total amount of SNAP-25 [F(2,12) = 0.06, *p* = 0.94] and synapsin I [F(2,12) = 0.07, *p* = 0.93]. We also showed that 10 μM albiflorin reduced the 4-AP-evoked increase in p-138-SNAP-25 and p-S9-synapsin I level (*p* < 0.001).

### 2.5. Albiflorin Reduces the Release Competence of Synaptic Vesicles Evoked by 4-AP in Synaptosomes

The regulation of synaptic vesicles by albiflorin was further evaluated using transmission electron microscopy (TEM). As shown in Figure 6A,B, the synaptosome contained numerous synaptic vesicles (control group). Synaptosomes treated with 1 mM 4-AP showed significantly fewer synaptic vesicles than control synaptosomes [F(2,6) = 65.3, *p* < 0.001], indicating that 4-AP evokes the release of all released vesicles. However, the number of synaptic vesicles was greater in synaptosomes preincubated with 10 μM albiflorin for 10 min before 4-AP application than in those treated with 4-AP alone (*p* < 0.001).

## 3. Discussion

This work focused on evaluating the effects of albiflorin on the release of glutamate from synaptosomes acutely prepared from the cerebral cortex of adult rats. We report here that albiflorin reduced vesicular exocytotic glutamate release from cerebrocortical synaptosomes via the depression of Ca^2+^ channel activation and the PKA pathway.

### 3.1. The Mechanism by Which Albiflorin Inhibited the 4-AP-Evoked Glutamate Release

The K^+^ channel blocker 4-AP, which mimics the physiological mechanisms of terminal depolarization and the Ca^2+^-dependent release of neurotransmitters, is considered a quasi-physiological stimulus for investigating the characteristics of the induced release of glutamate [23]. In the present study, 1 mM 4-AP elicited the release of a large amount of glutamate from nerve terminals, consistent with previous studies [24,25]. Albiflorin reduced this 4-AP-elicited glutamate release in a dose-dependent manner (3–30 μM) and reached a maximum effect at 18 μM. Furthermore, the albiflorin-mediated decrease in glutamate release was abolished by a blockade of vesicular glutamate transporters. This suggests that the albiflorin-mediated inhibition of 4-AP-elicited glutamate release is mediated by a decrease in the exocytotic pool available for release. The mechanisms possibly involved in the ability of albiflorin to inhibit vesicular exocytotic glutamate release were further assessed. In particular, the involvement of voltage-dependent Ca^2+^ channels, which are primarily contributed to the activation of vesicle exocytosis, was investigated [26]. Ca^2+^ channels, which are primarily supported for glutamate exocytosis, are N-type and P/Q-type [19,27,28]. Notably, we found that the depression of P/Q-type Ca^2+^ channels was involved in the glutamate-release-inhibiting effect of albiflorin, as indicated by the blockage of the action of albiflorin by the P/Q-type Ca^2+^ channel inhibitor, while the N-type Ca^2+^ channel inhibitor was ineffective. The glutamate release measured in the presence of both ω-agatoxin IVA and albiflorin was not significantly different from that observed in the presence of ω-agatoxin IVA alone. The lack of additivity in the actions of albiflorin and ω-agatoxin IVA can be explained by the inhibition of the same release pathway by both compounds. Moreover, the effect of albiflorin on 4-AP-elicited glutamate release was also significantly affected in the absence of extracellular Ca^2+^. These results suggest that albiflorin depresses vesicular exocytotic glutamate release from synaptosomes by decreasing extracellular Ca^2+^ entry via P/Q-type Ca^2+^ channels.

Additionally, the inhibition of Na^+^ channels or activation of K^+^ channels causes presynaptic inhibition resulting from the hyperpolarization of nerve terminals. This results in a reduction in Ca^2+^ influx and a consequent decrease in neurotransmitter release [29,30]. This mechanism did not involve the inhibitory effect of 4-AP-elicited glutamate release by albiflorin observed in the present study. This notion is supported by our observation that albiflorin did not influence 4-AP-evoked plasma membrane depolarization in synaptosomes. Although no direct evidence has indicated that albiflorin acts on presynaptic Ca^2+^ channels, our tentative conclusion is that the inhibitory effect of glutamate release by albiflorin occurs primarily through direct regulation of Ca^2+^ channels to affect Ca^2+^ entry. Further investigations are needed to better understand how albiflorin leads to the suppression of Ca^2+^ channels.

In glutamatergic nerve terminals, PKA increases glutamate release, and the phosphorylation of synaptic proteins in the release machinery, such as SNAP-25 and synapsin I, could be involved [20]. SNAP-25 and synapsin I are phosphorylated by PKA at T138 and S9, respectively. This phosphorylation promotes synaptic vesicle trafficking and increases the size of the releasable vesicle pools [22,31]. In the present study, when albiflorin was combined with the PKA inhibitor H89, the inhibitory effect of 4-AP-elicited glutamate release observed was not different from that observed with H89 alone. The lack of synergy in the effects of albiflorin and H89 can be explained by the notion that both factors inhibit the same pathway. In addition, we also observed increased levels of p-PKA, p-138-SNAP-25, and p-S9-synapsin I in 4-AP-treated cortical synaptosomes, which may involve a 4-AP-induced increase in the number of releasable vesicles and therefore increasing glutamate exocytosis. Furthermore, we found that albiflorin significantly decreased the levels of p-PKA, p-T138-SNAP-25, and p-S9-synapsin I in 4-AP-treated cortical synaptosomes. Therefore, it seems that the suppression of PKA by albiflorin reduces the phosphorylation of synapsin I and SNAP-25, resulting in attenuation of the number of releasable synaptic vesicles and reduced glutamate release. Consistent with this hypothesis, TEM revealed that 4-AP-induced synaptic vesicle changes were significantly reduced in albiflorin-treated cortical synaptosomes.

### 3.2. Therapeutic Implications

Albiflorin has no obvious acute oral toxicity and is a safe natural drug ingredient [7]. The ability of albiflorin to inhibit glutamate release from nerve terminals in the present study is of special interest, considering that excess glutamate is involved in many brain disorders, including neurodegenerative diseases (e.g., Alzheimer’s disease and Parkinson’s disease), and in mood disorders (e.g., depression) [32,33]. The reduction in glutamate release by albiflorin needs further investigation, and this observation represents a crucial mechanism of action of albiflorin that might contribute to its therapeutic effect on brain disorders. We infer that the glutamate-release-inhibiting effects of albiflorin might contribute to the reported effects on Alzheimer’s disease and depression [9,11,34].

## 4. Materials and Methods

### 4.1. Animals

Male Sprague Dawley rats weighing 180–200 g were purchased from the BioLASCO Taiwan Co., Ltd. (Taipei, Taiwan). The animal care and treatment were conducted in accordance with the Guide for the Care and Use of Laboratory Animals of the National Institutes of Health. The experimental protocol was approved by the Fu Jen Catholic University Animal Care Committee (A11208). In brief, the animals were kept in a room with controlled environmental conditions, maintaining a steady temperature of 23 ± 3 °C, a relative humidity of 50 ± 10%, and a 12 h light/dark cycle. The animals were provided with a standard rodent diet and had unrestricted access to purified water. All efforts were made to minimize the number of animals used and to reduce their suffering. A total of 25 animals were utilized in the study.

### 4.2. Preparation of Synaptosomes

Synaptosomes were isolated using Percoll density gradient centrifugation [24]. The experimental rats were sacrificed by decapitation. Their brains were quickly extracted and placed in a chilled medium containing 320 mM sucrose with a pH of 7.4. The cerebral cortex was then dissected and homogenized in the sucrose medium. The homogenate was centrifuged at 3000× *g* for 2 min at 4 °C. The resulting supernatant was then centrifuged again at 14,500× *g* for 12 min at 4 °C. The pellet was resuspended in a HEPES-buffered medium. The suspension was then separated using a layered, discontinuous Percoll gradient, which contained 320 mM sucrose, 1 mM EDTA, 0.25 mM dl-dithiothreitol, and Percoll concentrations of 3%, 10%, and 23%. The gradients were then centrifuged at 32,500× *g* for 7 min at 4 °C. Synaptosomes were recovered from the 10 and 23% Percoll bands and diluted in a HEPES buffer medium (HBM). This medium contained 140 mM NaCl, 5 mM KCl, 5 mM NaHCO_3_, 1 mM MgCl_2_·6H_2_O, 1.2 mM Na_2_HPO_4_, 10 mM glucose, and 10 mM HEPES, with a pH of 7.4. Following further centrifugation at 27,000× *g* for 10 min at 4 °C, the synaptosomal pellets were resuspended in HBM, and the protein content was determined by the Bradford assay (Thermo Fisher Scientific, Waltham, MA, USA). A volume of 0.5 mg of synaptosomal suspension was diluted in HBM and centrifugated (3000× *g* at 4 °C) for 10 min. The synaptosomal pellet was divided into three independent fractions used for the following purposes: (a) analysis of the glutamate release, plasma membrane potential, and Na^+^ concentration; (b) evaluation of the expression of p-PKA, p-T138-SNAP-25, and p-S9-synapsin I by Western blot; and (c) evaluation of the changes in synaptic vesicles by TEM.

### 4.3. Glutamate Release Assay

The release of glutamate from purified cerebrocortical synaptosomes was monitored in real time using an assay. Exogenous glutamate dehydrogenase (GDH) and NADP^+^ were used in this assay to couple the oxidative deamination of the released glutamate to the generation of NADPH. This process was detected using fluorometric methods [35,36]. Synaptosomal pellets were resuspended in HBM containing 16 M bovine serum albumin to bind any free fatty acids. A 2 mL aliquot was transferred to a stirred, thermostatted cuvette at 37 °C and incubated in the presence of 1 mM NADP^+^, 50 units of glutamate dehydrogenase, and 1.2 mM CaCl_2_. The released glutamate undergoes oxidative deamination by GDH to produce NADPH, resulting in an increase in fluorescence. This increase in fluorescence, with excitation and emission wavelengths of 340 and 460 nm, respectively, was measured to monitor glutamate release. The fluorescence of NADPH was detected using LS-55B (PerkinElmer, Waltham, MA, USA) model spectrofluorimeter. At the end of each experiment, 5 nmol of exogenous glutamate was added as a standard. Data were obtained at 2 s intervals. The fluorescence response used to quantify the released glutamate after a 5 min depolarization with 1 mM 4-AP was expressed as nanomoles of glutamate per milligram of synaptosomal protein per 5 min (nmol/mg/5 min).

### 4.4. Plasma Membrane Potential

The plasma membrane potential was evaluated using DiSC3(5), a dye that is advantageous due to its ability to alter fluorescence intensity in response to changes in membrane potential [37]. Synaptosomes were preincubated and resuspended following the protocol used for the glutamate release experiments. After 3 min of incubation, 5 μM DiSC3(5) was added and allowed to equilibrate. Following 4 min of incubation, CaCl_2_ (1.2 mM) was introduced. At the 10 min mark, 4-AP (1 mM) was added to depolarize the synaptosomes. The fluorescence of DiSC3(5) was then monitored at excitation and emission wavelengths of 646 nm and 674 nm, respectively. The results were expressed in fluorescence units.

### 4.5. TEM

For ultrastructural analysis, purified cortical synaptosomes were fixed in 4% paraformaldehyde and 0.1% glutaraldehyde for 12 h at 4 °C. The fixed cortical synaptosomes were rinsed in phosphate-buffered saline (PBS), then post-fixed in 1% osmium tetroxide for 2 h. Afterward, they were dehydrated and embedded in epoxy resin. Next, ultrathin sections were cut to a thickness of 70 nm using an ultramicrotome (EM UC7, Leica Microsystems, Wetzlar, Germany). A transmission electron microscope (JEM-1400, JEOL, Tokyo, Japan) was used to examine the ultrastructure of the cortical synaptosomes.

### 4.6. Western Blotting

The method and conditions for preparing samples for Western blot analysis have been described previously [38]. Synaptosomal lysates (20 mg/lane) were subjected to sodium dodecylsulfate polyacrylamide gel electrophoresis (SDS-PAGE) in 10% gel under reducing conditions and electroblotted onto polyvinylidene fluoride (PVDF) membranes (Bio-Rad Laboratories, Hercules, CA, USA). After electroblotting, non-specific binding sites in membranes were blocked with Tris-buffered saline containing Tween-20 (TBS-T: 20 mM Tris, 500 mM NaCl, 0.1% Tween 20, pH 7.5) and 5% skimmed milk at room temperature for 1 h. Subsequently, membranes were incubated overnight at 4 °C with the primary antibody. Membranes were washed with TBS-T and incubated for 2 h with horseradish peroxidase–coupled secondary antibody at room temperature. Enhanced chemiluminescence (Amersham, Buckinghamshire, UK) was used to visualize the protein bands, which were exposed to X-ray film. Films were scanned, and the level of proteins was assessed using the Image J software (version 1.45 J, National Institutes of Health, Bethesda, Rockville, MD, USA). The density of the band of interest was normalized against β-actin to determine the relative band density ratio. The primary antibodies used in this study included β-actin (1:5000), PKA (1:2000), p-PKA (1:2000), SNAP-25 (1:50,000), p-T138-SNAP-25 (1:500), synapsin I (1:50,000), and p-S9-synapsin I (1:700). β-actin, PKA, p-PKA, synapsin I, and p-S9-synapsin I were purchased from Cell Signaling (Danvers, MA, USA). SNAP-25 was obtained from Abcam (Cambridge, UK). p-T138-SNAP-25 was obtained from Aviva Systems Biology Corporation (San Diego, CA, USA).

### 4.7. Data Analysis

The Shapiro–Wilk test was used to test the data normality. Data analysis and graph creation were performed using GraphPad Prism version 8.0 (GraphPad Software, San Diego, CA, USA). The results were analyzed using one-way analysis of variance (ANOVA), followed by Tukey’s post hoc test when comparing more than two groups. A two-tailed Student’s *t*-test was used to determine whether there was a significant difference between the means of two groups. Data were presented as mean ± standard error of the mean (SEM). The statistical significance was set at *p* < 0.05.

### 4.8. Chemicals

Albiflorin (purity > 98%) was purchased from ChemFaces (Wuhan, Hubei, China). The calcium channel blockers ω-conotoxin GVIA and ω-agatoxin IVA were purchased from Alomone (Jerusalem, Israel). DiSC_3_(5) was acquired from Invitrogen (Carlsbad, CA, USA). Bafilomycin A1 and KT5720 were obtained from Tocris Cookson (Bristol, UK). 4-AP, H89, Percoll, sucrose, EDTA, dl-dithiothreitol, HEPES, GDH, NADP^+^, SDS and general reagents were acquired from Sigma-Aldrich (St. Louis, MO, USA). Albiflorin, DiSC_3_(5), and H89 were dissolved in 0.1% (14.3 mM) dimethylsulfoxide (DMSO). 4-AP, ω-conotoxin GVIA, and ω-agatoxin IVA were dissolved in distilled water.

## 5. Conclusions

Our data demonstrated that by decreasing Ca^2+^ entry via P/Q-type Ca^2+^ channels and reducing PKA-mediated synapsin I and SNAP-25 phosphorylation, albiflorin inhibits vesicular exocytotic glutamate release from rat cerebral cortical synaptosomes. This investigation enhances the understanding of albiflorin action in the brain and suggests that albiflorin is valuable for treating glutamate-induced neurological disorders.

## Figures and Tables

**Figure 1 ijms-25-08846-f001:**
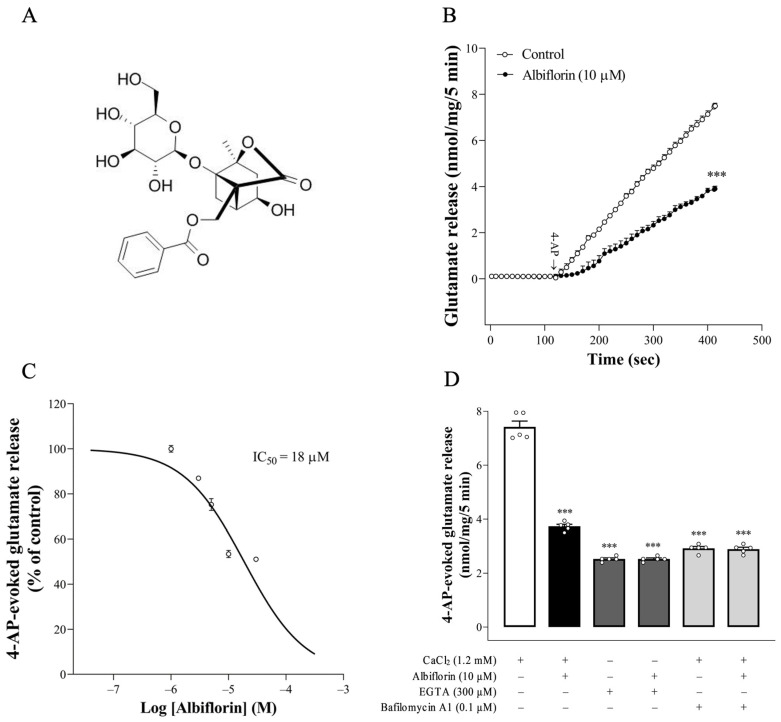
Albiflorin suppresses the release of glutamate induced by 4-AP from rat cerebrocortical nerve terminals. (**A**) Albiflorin molecular structure. (**B**) Glutamate release triggered by 4-AP (1 mM) in both the control condition and in the presence of albiflorin. (**C**) Dose–response curves of 4-AP-evoked glutamate release for the calculation of IC50 values. The concentration of albiflorin ranges from 1 to 30 μM. (**D**) Effects of Ca^2+^-free medium containing 0.3 mM EGTA, 0.2 μM bafilomycin A1, and 10 μM albiflorin alone or concomitantly added on the 4-AP (1 mM)-elicited glutamate release. Albiflorin or bafilomycin A1 was added 10 min before 4-AP addition. Results are the means ± SEM (n = 5 rats/group). *** *p* < 0.001 versus control group.

**Figure 2 ijms-25-08846-f002:**
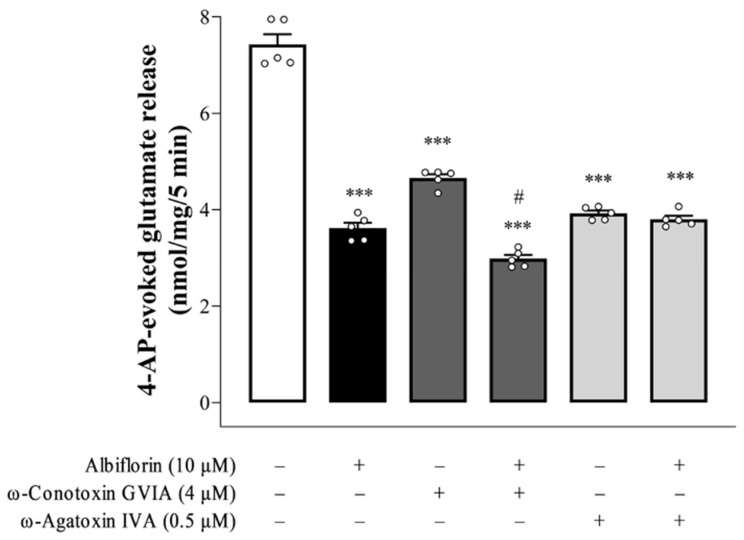
The inhibition of 4-AP-induced glutamate release by albiflorin is blocked by the P/Q-type Ca^2+^ channel inhibitor ω-agatoxin IVA but not by the N-type Ca^2+^ channel inhibitor ω-conotoxin GVIA. Effects of neuronal voltage-sensitive Ca^2+^ channel blockers, ω-conotoxin GVIA or ω-agatoxin IVA, in combination with or without albiflorin, on the 1 mM 4-AP-elicited glutamate release. Albiflorin, ω-conotoxin GVIA or ω-agatoxin IVA was added 10 min before the addition of 4-AP. Results are the means ± SEM (n = 5 rats/group). *** *p* < 0.001 versus control group, # *p* < 0.001 versus ω-conotoxin GVIA-treated group.

**Figure 3 ijms-25-08846-f003:**
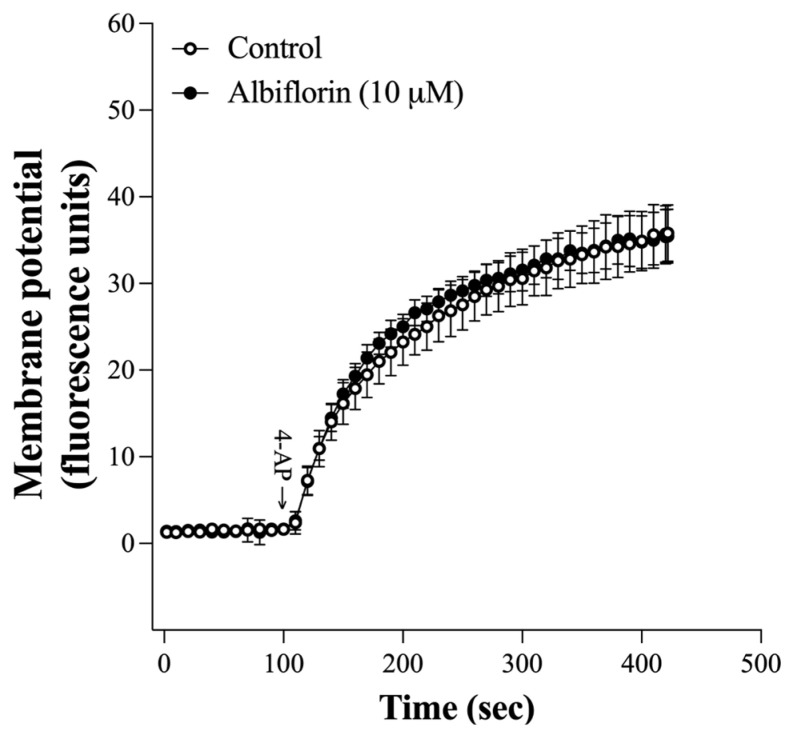
Albiflorin does not affect the synaptosomal membrane potential. The membrane potential of synaptosomes was measured using DiSC3(5) during depolarization with 4-AP (1 mM), either without (control) or with albiflorin, which was added 10 min prior to the 4-AP. Results are the means ± SEM (n = 5 rats/group).

**Figure 4 ijms-25-08846-f004:**
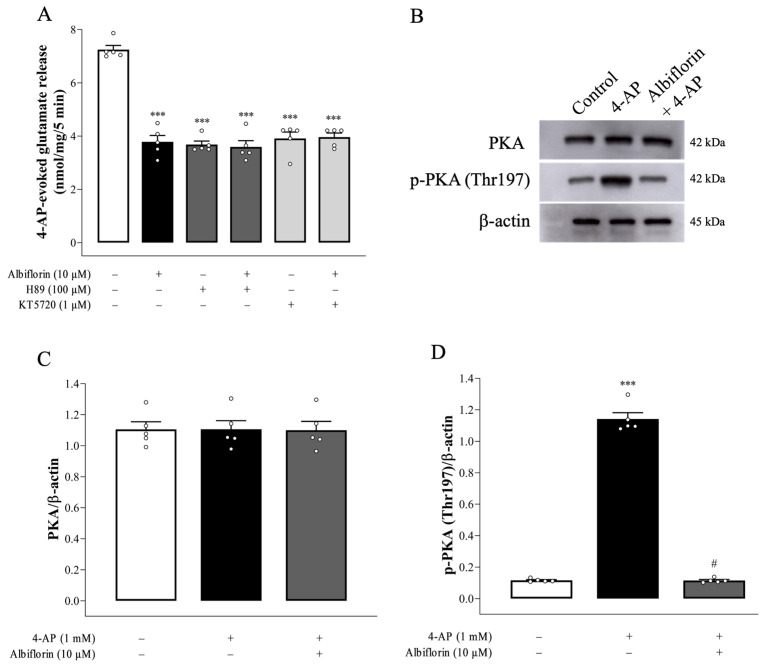
Albiflorin-mediated inhibition of 4-AP-evoked glutamate release is prevented by PKA inhibitors H89 and KT5720. (**A**) Effects of H89, KT5720, and albiflorin alone or concomitantly added on the 4-AP (1 mM)-elicited glutamate release. (**B**) Representative images and (**C**,**D**) quantification of PKA and p-Thr197-PKA protein levels in the absence (control) or presence of 4-AP or 4-AP + albiflorin. H89 and KT5720 were added 30 min before 4-AP addition. Results are the means ± SEM (n = 5 rats/group). *** *p* < 0.001 versus control group, # *p* < 0.001 versus 4-AP-treated group.

**Figure 5 ijms-25-08846-f005:**
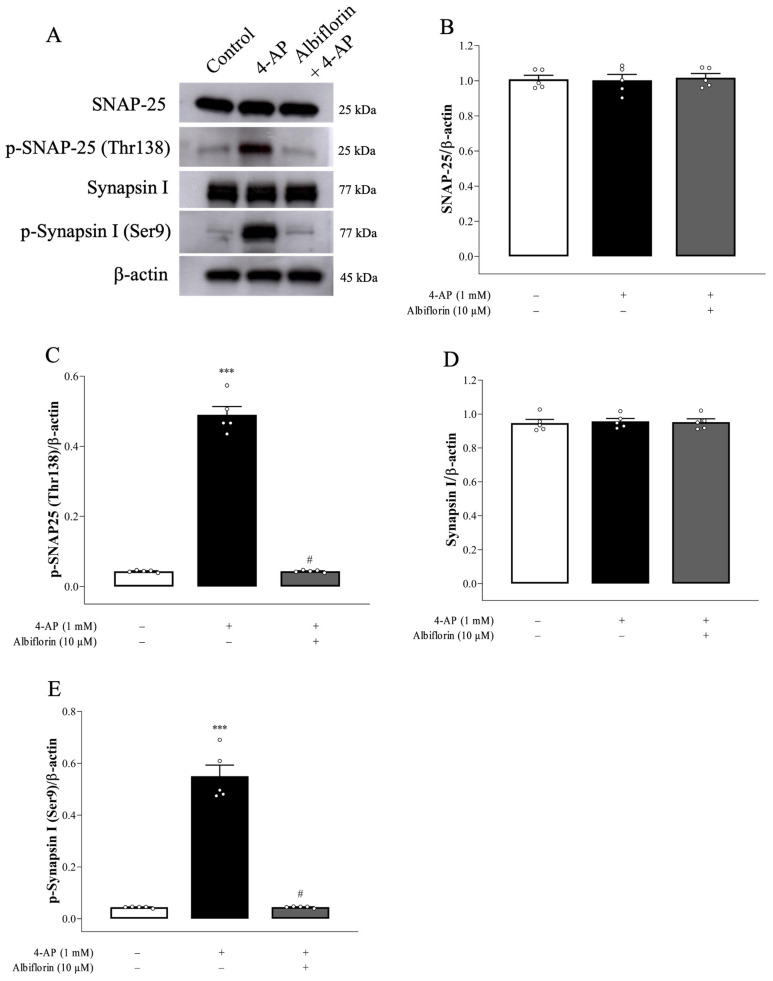
Albiflorin decreases 4-AP-elicited phosphorylation of synapsin I and SNAP-25. (**A**) Representative images and (**B**–**E**) quantification of synapsin-1, p-Ser9-synapsin I, SNAP-25, and p-T138-SNAP-25 protein levels in the absence (control) or presence of 4-AP or 4-AP + albiflorin. Albiflorin was added 10 min before 4-AP addition. Results are the means ± SEM (n = 5 rats/group). *** *p* < 0.001 versus control group, # *p* < 0.001 versus 4-AP-treated group.

**Figure 6 ijms-25-08846-f006:**
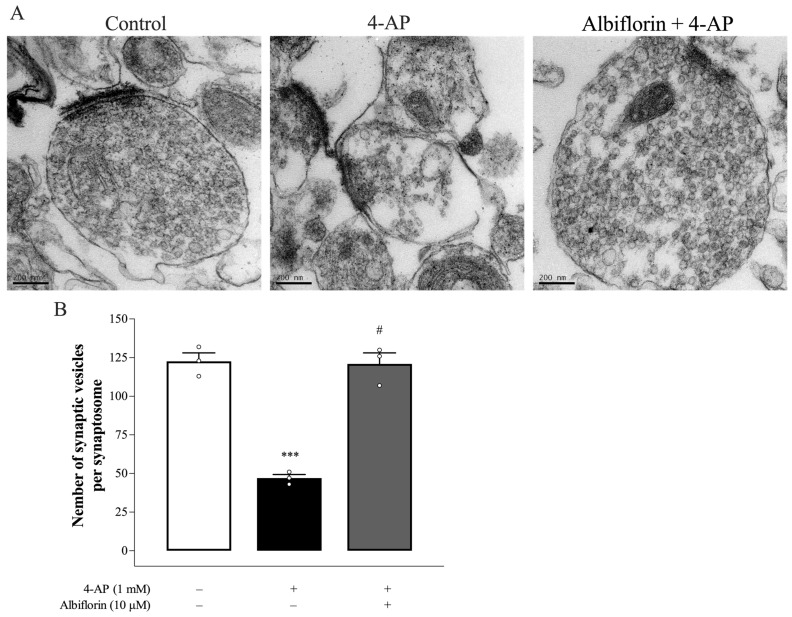
The effect of albiflorin on 4-AP-evoked changes in synaptic vesicles in synaptosomes. The synapse ultrastructure (**A**) and the number of synaptic vesicles (**B**) in each synaptosome were observed using TEM. Scale bar: 200 nm, magnification: 60,000×. Results are the means ± SEM (n = 3 rats/group). *** *p* < 0.001 versus control group, # *p* < 0.001 versus 4-AP-treated group.

## Data Availability

Data will be made available on request.

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
