# Peer review of "Albiflorin Decreases Glutamate Release from Rat Cerebral Cortex Nerve Terminals (Synaptosomes) through Depressing P/Q-Type Calcium Channels and Protein Kinase A Activity"

_ijms, 2024, doi:10.3390/ijms25168846_

Round 1

Reviewer 1 Report

Comments and Suggestions for Authors

The authors present a complete characterization of possible action mechanism of albiflorin, a natural monoterpene glycoside, that affects the release of glutamate, using cutting-edge techniques. 

1.- The work performed proportionate information about the effect of albiflorin on the regulation of glutamate release, a mechanism related with the neuroprotective action described for this compound. This issue had not been studied previously in literature, which makes this work an original and novel proposal.    2.- The experiments realized are enough and described sufficiently in the materials and methods section. Furthermore, as mentioned previously, the technical quality is state-of-the-art (the use of synaptosomes as model, on-line fluorimetry, transmission electron microscopy (TEM), plasma membrane potential, and western blotting),  providing clear and convincing data. All of them in the context of the question to be answered (the effect of albiflorin on the regulation of glutamate release).   3.- Finally, the work contributed significantly in the area of central nervous system (CNS) disease, not only providing information about the action mechanism of albiflorin, but also suggesting a methodological strategy to study molecules that act at the level of CNS.

The results reported have scientific soundness and contribute to understand the weather this compound acts in the brain. These data will be useful to those involved in brain related diseases research. Therefore, I recommend its publication in its actual form.

Author Response

Reviewer 1

The authors present a complete characterization of possible action mechanism of albiflorin, a natural monoterpene glycoside, that affects the release of glutamate, using cutting-edge techniques.

1.- The work performed proportionate information about the effect of albiflorin on the regulation of glutamate release, a mechanism related with the neuroprotective action described for this compound. This issue had not been studied previously in literature, which makes this work an original and novel proposal.    2.- The experiments realized are enough and described sufficiently in the materials and methods section. Furthermore, as mentioned previously, the technical quality is state-of-the-art (the use of synaptosomes as model, on-line fluorimetry, transmission electron microscopy (TEM), plasma membrane potential, and western blotting),  providing clear and convincing data. All of them in the context of the question to be answered (the effect of albiflorin on the regulation of glutamate release).   3.- Finally, the work contributed significantly in the area of central nervous system (CNS) disease, not only providing information about the action mechanism of albiflorin, but also suggesting a methodological strategy to study molecules that act at the level of CNS.

The results reported have scientific soundness and contribute to understand the weather this compound acts in the brain. These data will be useful to those involved in brain related diseases research. Therefore, I recommend its publication in its actual form.

Thank you to the review committee for the affirmation.

Reviewer 2 Report

Comments and Suggestions for Authors

Lu et al:

The authors present a study where they investigate glutamate release from synaptosomes. They see that 4-AP-elicited release of glutamate was attenuated by the compound albiflorin. They see this inhibition with Calcium channel inhibitors as well as inhibitors of glutamate vesicular transport, with and without albiflorin. They also show that albiflorin reduced the phosphorylation of PKA and its downstream targets that affect glutamate release.

Major comments:

I like the paper overall but do not understand the rationale for using Calcium channel blockers (conotoxin/agatoxin) and glutamate vesicular transport inhibitor (bafilomycin). I understand that the mechanism of release may involve Calcium release and glutamate transport, it likely does. However, hitting the same pathway twice does not necessarily mean that albiflorin acts via these pathways. Unless these are the only pathways of vesicular release of glutamate (in which case there is no need to do this), the authors should maybe show how albiflorin may attenuate an increase in glutamate secretion by activation of such pathways. Meaning if these pathways (calcium release and glutamate transport) are activated, and lead to increase glutamate secretion, does albiflorin bring down such an increase?

They say in their discussion:

"Furthermore, the albiflorin-mediated inhibition of glutamate release was prevented by blockade of vesicular glutamate transporters"- I would tend to disagree. That would mean that albiflorin-mediated decrease was reversed by these blockades, but the effects are very similar. Similarly, they also state that:

"Furthermore, the inhibitory effect of albiflorin on 4-AP-evoked glutamate release was also prevented by the absence of extracellular Ca2+."

I found both these statements to be over-interpretations. Absence of extracellular Calcium and application albiflorin have the same effect, just like the previous quoted statement. So, how is there a prevention of albiflorin's inhibitory effect?

Minor comment:

Figure 5A is missing lane descriptions.

Author Response

Major comments:

I like the paper overall but do not understand the rationale for using Calcium channel blockers (conotoxin/agatoxin) and glutamate vesicular transport inhibitor (bafilomycin). I understand that the mechanism of release may involve Calcium release and glutamate transport, it likely does. However, hitting the same pathway twice does not necessarily mean that albiflorin acts via these pathways. Unless these are the only pathways of vesicular release of glutamate (in which case there is no need to do this), the authors should maybe show how albiflorin may attenuate an increase in glutamate secretion by activation of such pathways. Meaning if these pathways (calcium release and glutamate transport) are activated, and lead to increase glutamate secretion, does albiflorin bring down such an increase?

They say in their discussion:

"Furthermore, the albiflorin-mediated inhibition of glutamate release was prevented by blockade of vesicular glutamate transporters"- I would tend to disagree. That would mean that albiflorin-mediated decrease was reversed by these blockades, but the effects are very similar. Similarly, they also state that:

"Furthermore, the inhibitory effect of albiflorin on 4-AP-evoked glutamate release was also prevented by the absence of extracellular Ca2+."

I found both these statements to be over-interpretations. Absence of extracellular Calcium and application albiflorin have the same effect, just like the previous quoted statement. So, how is there a prevention of albiflorin's inhibitory effect?

We thank the reviewer for the critical comments and constructive suggestions.

In the present study, to investigate the effect of albiflorin on glutamate release, isolated nerve terminals were depolarized with the K+ channel blocker 4-AP. 4-AP is a K+ channel blocker that mimics the physiological mechanisms of terminal depolarization and the Ca2+-dependent vesicular exocytotic glutamate release. Therefore, the effect of albiflorin on evoked glutamate release was examined in the absence of extracellular Ca2+ or in the presence of vesicular glutamate transporter inhibitor bafilomycin or Ca2+ channel blockers. Ca2+-free medium, bafilomycin A1 and conotoxin/agatoxin significantly affected glutamate efflux elicited by 4-AP. However, no statistical difference was observed between the release after bafilomycin A1, Ca2+ free medium or agatoxin alone and after bafilomycin A1 + albiflorin, Ca2+ free medium + albiflorin or agatoxin + albiflorin treatment. These results indicate that albiflorin reduces Ca2+-dependent vesicular exocytotic glutamate release from cerebral cortical synaptosomes in rats. In order to make the clear of statement, the sentences are modified (Page 2, lines 71-73, 87-89; Page 9, lines 196-198, 206-210, 220-224).

Minor comment:

Figure 5A is missing lane descriptions.

Figure 5A is modified.

Reviewer 3 Report

Comments and Suggestions for Authors

The manuscript by Lu et al. “Albiflorin inhibits glutamate release from rat cerebrocortical nerve terminals (synaptosomes) by suppressing P/Q-type calcium channels and the protein kinase A pathways” elucidates the suppression effect of albiflorin on 4-AP-induced glutamate release from synatosomes prepared from rat cerebral cortex. In this study, the mechanisms underlying albiflorin-suppressed vesicular exocytotic glutamate release are proposed that albiflorin reduces Ca2+ influx through P/Q-type Ca2+ channels and phosphorylation of SNAP-25 and synapsin I via PKA. Employing excellent biochemical techniques and transmission electron microscopy, authors demonstrated neuroprotective action of albiflorin with beautiful data. The manuscript is described logically and easy to follow authors’ argument.

Albeit the experiments are overall well designed and carried out, some concerns on ambiguity of concepts and insufficient of quantitative analyses remain to be addressed before this manuscript is suitable for publication. Particularly, the curve-fitting model of albiflorin concentration to reduction of 4-AP-evoked glutamate release shown in figure 1C is contradiction to results shown in figure 1D. The curve fit assumes that glutamate release goes to zero. But, as shown in figure 1D, glutamate release did not decrease below 2 nmol/mg/5 min (~25% of control) even in the concomitant of albiflorin and barifomycin A1 or EGTA. The lack of an additional decrease in glutamate release by albiflorin implies saturation of inhibition to vesicular exocytotic glutamate release. So, 4-AP-evoked glutamate release in control includes an insensitive component to albiflorin. It should be corrected the model for curve fitting (Fig. 1C) to a model composed by sensitive and insensitive components to albiflorin.

(1)    Line 43. It would be more helpful to readers to cite references for relationships between excitotoxicity and brain diseases, including stroke, traumatic brain injury, and Alzheimer’s disease.

(2)    Line 73. It would be kind to readers to mention a role of 4-AP, a blocker of voltage-gated K+ channels, to cause depolarization in synaptosomal membrane, at first.

(3)    Line 77. Is it possible that pretreatment with Albiflorin for 10 min is depleting intracellular glutamate before 4-AP application?

(4)    Line 78. Although the description of t (8) indicates 8 for degree of freedom of t-test, it is also better to show specific n number of control and albiflorin in the figure legend of 1B.

(5)    Line 81. For the unit of the vertical axis of the graph in Figure 1C, it is supposed to be “%” rather than “nmol/mg/5 min”. Furthermore, the IC50 value will change for the reasons mentioned above.

(6)    Line 83. The description of F (2, 12) indicates groups of 3 and n of 15 total for testing ANOVA. Which 3 of 6 groups shown in 1D were tested with ANOVA? In addition, I believe it is necessary to mention here the p of the post hoc test between the use of bafilomycin A1 alone and its combination with albiflorin. Clear statement of statistics is needed throughout the manuscript.

(7)    Line 109 and 111. Correction of “w”-agatoxin IVA to symbol font is needed.

(8)    Line 113. Please mention what the symbol "#" in Figure 2 indicates.

(9)    Line 117. "Albiforin" seems to be a misspelling of "albiflorin."

(10)    Line 126. In figure 3, there is no graph of (B). Authors have deleted results of Na+ influx in the manuscript while related description remains in legend of figure 3 and the section of material and methods (line 288 “Na+ concentration” and line 358 “SBFI-AM”).

(11)    Line 169. Labels indicating the treatment condition is needed on top of figure 5A such as figure 4B.

(12)    Line 205. Authors wrote “...reached a maximum effect at 10 mM.”. It is inconsistent with the IC50 of 18 mM. For the reasons mentioned above, the IC50 value needs to be reconsidered.

(13)    Line 284. The abbreviation HBM is used without explanation.

(14)    Line 295. Correct “at 37 uC” to “at 37 °c”.

(15)    Line 327. Remove “n” from “by n 12%”.

(16)    Line 359. It would be better to clarify final concentration of DMSO.

Author Response

Reviewer 2

We thank the reviewer for the critical comments and constructive suggestions.

Albeit the experiments are overall well designed and carried out, some concerns on ambiguity of concepts and insufficient of quantitative analyses remain to be addressed before this manuscript is suitable for publication. Particularly, the curve-fitting model of albiflorin concentration to reduction of 4-AP-evoked glutamate release shown in figure 1C is contradiction to results shown in figure 1D. The curve fit assumes that glutamate release goes to zero. But, as shown in figure 1D, glutamate release did not decrease below 2 nmol/mg/5 min (~25% of control) even in the concomitant of albiflorin and barifomycin A1 or EGTA. The lack of an additional decrease in glutamate release by albiflorin implies saturation of inhibition to vesicular exocytotic glutamate release. So, 4-AP-evoked glutamate release in control includes an insensitive component to albiflorin. It should be corrected the model for curve fitting (Fig. 1C) to a model composed by sensitive and insensitive components to albiflorin.

As suggestion by the reviewer, Fig. 1 C is modified.

(1)    Line 43. It would be more helpful to readers to cite references for relationships between excitotoxicity and brain diseases, including stroke, traumatic brain injury, and Alzheimer’s disease.

The reference is cited [Line 43].

(2)    Line 73. It would be kind to readers to mention a role of 4-AP, a blocker of voltage-gated K+ channels, to cause depolarization in synaptosomal membrane, at first.

The sentences [The K+-channel blocker 4-AP destabilizes the membrane potential and is thought to cause repetitive spontaneous Na+-channel-dependent depolarization that closely approximates in vivo depolarization of the synaptic terminal, which leads to the activation of volt-age-dependent Ca2+ channels (VDCCs) and neurotransmitter release (Nicholls, 1998)] are added in the result section (Line 74-78).

(3)    Line 77. Is it possible that pretreatment with Albiflorin for 10 min is depleting intracellular glutamate before 4-AP application?

About this point, the sentence [Albiflorin did not alter the basal release of glutamate (p > 0.05)] is added in the result section (Line 82-83).

(4)    Line 78. Although the description of t (8) indicates 8 for degree of freedom of t-test, it is also better to show specific n number of control and albiflorin in the figure legend of 1B.

n = 5 rats/group is included in the legend of Fig. 1 (Line 102).

(5)    Line 81. For the unit of the vertical axis of the graph in Figure 1C, it is supposed to be “%” rather than “nmol/mg/5 min”. Furthermore, the IC50 value will change for the reasons mentioned above.

As suggestion by the reviewer, Fig. 1 C is modified.

(6)    Line 83. The description of F (2, 12) indicates groups of 3 and n of 15 total for testing ANOVA. Which 3 of 6 groups shown in 1D were tested with ANOVA? In addition, I believe it is necessary to mention here the p of the post hoc test between the use of bafilomycin A1 alone and its combination with albiflorin. Clear statement of statistics is needed throughout the manuscript.

In this study, comparisons among multiple groups were analyzed by one-way analysis of variance (ANOVA) and post hoc Tukey’s tests. In order to make the clear statement, the sentences are modified (Lines 92-93).

(7)    Line 109 and 111. Correction of “w”-agatoxin IVA to symbol font is needed.

The words are corrected (Line 115, 117).

(8)    Line 113. Please mention what the symbol "#" in Figure 2 indicates.

The sentence [# p < 0.001 versus w-conotoxin GVIA-treated group] is added (Lines 119-120).

(9)    Line 117. "Albiforin" seems to be a misspelling of "albiflorin."

The word is corrected (Line 124).

(10)    Line 126. In figure 3, there is no graph of (B). Authors have deleted results of Na+ influx in the manuscript while related description remains in legend of figure 3 and the section of material and methods (line 288 “Na+ concentration” and line 358 “SBFI-AM”).

The sentences are modified (Lines 133-137).

(11)    Line 169. Labels indicating the treatment condition is needed on top of figure 5A such as figure 4B.

Fig. 5A is modified.

(12)    Line 205. Authors wrote “...reached a maximum effect at 10 mM.”. It is inconsistent with the IC50 of 18 mM. For the reasons mentioned above, the IC50 value needs to be reconsidered.

10 mM is changed to 18 mM (Line 212).

(13)    Line 284. The abbreviation HBM is used without explanation.

The sentence is modified (Line 288).

(14)    Line 295. Correct “at 37 uC” to “at 37 °c”.

The word is corrected (Line 350).

(15)    Line 327. Remove “n” from “by n 12%”.

The word is corrected (Line 334).

(16)    Line 359. It would be better to clarify final concentration of DMSO.

0.1 % (14.3 mM) DMSO is added in the method section (Line 367).

Round 2

Reviewer 2 Report

Comments and Suggestions for Authors

Line 216 on page 9: "The inhibitory action of albiflorin on 4-AP-elicited glutamate release was decreased from 48% to 3% in the presence of P/Q-type Ca 2+ channel inhibitor ω-agatoxin IVA." The bar graph in Fig2 does not suggest such a drastic decrease. or do the authors imply "another 3%" ? I do agree with the authors in the discussion that Albiflorin and agatoxin may be hitting a similar release mechanism, which leads to lack of additive effects.

On page 3 lines 113-114, language should be changed as well. The lack of additional inhibition with agatoxin suggests similar pathways between agatoxin and albiflorin.

Author Response

Reviewer 2

We thank the reviewer for the critical comments and constructive suggestions.

Line 216 on page 9: "The inhibitory action of albiflorin on 4-AP-elicited glutamate release was decreased from 48% to 3% in the presence of P/Q-type Ca 2+ channel inhibitor ω-agatoxin IVA." The bar graph in Fig2 does not suggest such a drastic decrease. or do the authors imply "another 3%" ? I do agree with the authors in the discussion that Albiflorin and agatoxin may be hitting a similar release mechanism, which leads to lack of additive effects.

As suggestion by the reviewer, the sentence is modified (Page 9 Line 209-213).

On page 3 lines 113-114, language should be changed as well. The lack of additional inhibition with agatoxin suggests similar pathways between agatoxin and albiflorin.

As suggestion by the reviewer, the sentence is modified (Page 3 Line 107-111).
